# Is There a Foreign *Accent* Effect on Moral Judgment?

**DOI:** 10.3390/brainsci11121631

**Published:** 2021-12-10

**Authors:** Alice Foucart, Susanne Brouwer

**Affiliations:** 1Centro de Investigación Nebrija en Cognición (CINC), Facultad de Lenguas y Educación, Universidad Nebrija, 28015 Madrid, Spain; 2Centre for Language Studies, Radboud University, 6525 Nijmegen, The Netherlands; susanne.brouwer@ru.nl

**Keywords:** foreign accent, foreign language effect, emotion processing, cognitive disfluency, psychological distance

## Abstract

Recent studies have shown that people make more utilitarian decisions when dealing with a moral dilemma in a foreign language than in their native language. Emotion, cognitive load, and psychological distance have been put forward as explanations for this foreign *language* effect. The question that arises is whether a similar effect would be observed when processing a dilemma in one’s own language but spoken by a foreign-accented speaker. Indeed, foreign-accented speech has been shown to modulate emotion processing, to disrupt processing fluency and to increase psychological distance due to social categorisation. We tested this hypothesis by presenting 435 participants with two moral dilemmas, the trolley dilemma and the footbridge dilemma online, either in a native accent or a foreign accent. In Experiment 1, 184 native Spanish speakers listened to the dilemmas in Spanish recorded by a native speaker, a British English or a Cameroonian native speaker. In Experiment 2, 251 Dutch native speakers listened to the dilemmas in Dutch in their native accent, in a British English, a Turkish, or in a French accent. Results showed an increase in utilitarian decisions for the Cameroonian- and French-accented speech compared to the Spanish or Dutch native accent, respectively. When collapsing all the speakers from the two experiments, a similar increase in the foreign accent condition compared with the native accent condition was observed. This study is the first demonstration of a foreign *accent* effect on moral judgements, and despite the variability in the effect across accents, the findings suggest that a foreign accent, like a foreign language, is a linguistic context that modulates (neuro)cognitive mechanisms, and consequently, impacts our behaviour. More research is needed to follow up on this exploratory study and to understand the influence of factors such as emotion reduction, cognitive load, psychological distance, and speaker’s idiosyncratic features on moral judgments.

## 1. Introduction

Recent studies have shown that the use of a foreign language modifies our decisions. For instance, when people are asked whether they would kill one person to save five, they are more likely to answer positively when asked in a foreign language than in their native language (e.g., [1,2]). The origin of this foreign language effect (FLe) is still unclear, but the factors that have been put forward to explain it are a reduction in emotion, an increase in cognitive load, and psychological distance provoked by a foreign language [3]. Following the dual-process account [4,5] according to which moral decision making is driven by an interaction between rational and emotional processes, a foreign language could reduce the weight of the emotional processes during decision making. Hence, the language we use can be viewed as a contextual aspect that modulates (neuro)cognitive mechanisms, and consequently, impacts our behaviour. The question that arises is whether our decisions would be similarly affected if these mechanisms were modulated by other conditions. To address this question, we investigated whether moral judgements are modified by another linguistic context that similarly affects emotion, cognitive load and psychological distance: foreign-accented speech. This question is particularly relevant in a globalised world, in which more and more people speak one or multiple foreign languages in addition to their native language. Hence, the number of people who speak with a foreign accent has increased steadily, and therefore, it is essential to understand the effect an accent may have on native speakers’ behaviour. To test the possibility of a foreign *accent* effect, we presented participants with the well-attested trolley and footbridge dilemmas [6,7], as in Costa et al.’s (2014) study, with two exceptions. We did not manipulate the language of presentation but the accent, native versus foreign, and we presented the dilemmas not in written but in auditory form [8,9,10].

As mentioned, one of the factors that has been proposed to account for the FLe is a reduction in emotion. Indeed, previous research has shown that the response to emotional linguistic stimuli in a foreign language is attenuated compared to the response in a native language [11,12]. This reduction is due to the fact that a native language is acquired in a naturalistic, emotionally rich environment, whereas a foreign language is usually acquired in a classroom with limited emotional experience [13,14]. A similar reflection was made by Hatzidaki and colleagues regarding emotion and foreign accents [15]. Using event-related potentials, they recorded the brain activity of Spanish native speakers in response to emotional and neutral words spoken in native or foreign accents. They observed that the neural response (late positive complex) to (positive) emotional words was reduced when words were processed in a foreign accent. To account for their results, they proposed that when the words of a language are learnt early in life, both the linguistic and extralinguistic information is stored in memory, including accent. Therefore, the retrieval of these words may be easier when processed in a native accent than in a foreign accent. In relation to emotional words, if they are related to an event experienced in a native-accent context, their link with the episodic memory may be stronger when retrieved in a similar accent, which may modulate the arousal provoked by these words. Thus, foreign-accented speech, like a foreign language, has been shown to affect emotion processing. If, as claimed, this is the reduction in emotion provoked by a foreign language that affects moral judgements, we should observe a similar effect with a foreign accent.

Another factor that has been suggested to contribute to the FLe is the cognitive load generated by the disfluency linked to the processing of a foreign language [1,16]. The disruption of cognitive fluency has been shown to reduce decision biases because it sets people in a more focused state of processing [17,18,19]. According to this explanation, when facing a moral dilemma, the cognitive load generated by the use of a foreign language would lead to more utilitarian responses. Alternatively, other studies have reported that taxing cognitive resources decreases utilitarian decisions [20,21], and recent evidence has demonstrated that using a foreign language does not always facilitate reasoning and may also sometimes hamper it [22,23]. Independently of whether it promotes or hampers reasoning, disfluency seems to affect our decisions, and foreign-accented speech, like the processing of a foreign language, disrupts fluency [24,25]. For instance, native speakers are less sensitive to syntactic errors produced by foreign than native speakers [26,27], anticipatory mechanisms are reduced in foreign-accented speech [28] and stories are remembered in less details in foreign compared to native speech [29]. In addition, the cognitive load generated by a foreign accent may lead to a negative bias towards the speaker [29,30,31,32,33]. This negative bias and the simple fact that a foreigner may be categorised as a member of a different social group provokes psychological distance [29,34]. An increase in psychological distance makes individuals construe situations in a more abstract ways, which may twist their decisions to utilitarianism [35], for example in the footbridge [1]. As, like a foreign language, foreign-accented speech disrupts fluency and increases psychological distance, the cognitive mechanisms that modulate decision making, we should observe a significant difference in utilitarian responses when moral judgements are presented in a foreign accent or in a native accent.

## 2. The Present Study

Data, analyses and model outputs will be available upon publication at Available online: https://osf.io/rey5w/ (accessed on 7 December 2021).

Here, we investigated whether foreign-accented speech affects the same cognitive mechanisms as when processing a foreign language. To test the hypothesis of a foreign *accent* effect, we presented participants with the well-attested trolley and footbridge dilemmas [6,7] in a native or in a foreign accent, in auditory form. The advantage of manipulating accent and not a foreign language, as in Costa et al. (2014), is that the dilemmas are presented in the participants’ native language in both conditions, which removes comprehension issues due to foreign language proficiency. To generalise the potential foreign *accent* effect, we conducted the study in two different native languages, Spanish (Experiments 1) and Dutch (Experiments 2).

We expected to replicate the foreign language effect with foreign-accented speech, i.e., we expected participants to be more likely to state they would kill one person to save five when facing a dilemma spoken in a foreign accent than in a native accent. Based on the FLe, we predicted this effect to occur on the footbridge dilemma only, since it is more emotionally charged than the trolley dilemma [4], in which no differences should be observed. Additionally, because the social categorization generated by a foreign accent may trigger stereotypes associated with the speaker’s culture [36,37], and because the accent strength and comprehensibility may vary across speakers, we used different speakers with various foreign accents, such as British English, Cameroonian, Turkish and French. However, given that the study was primarily designed to verify the possibility of a potential effect of accent on decisions, the factors that may modulate such effects (e.g., stereotypes, language attitudes) were not controlled for in this first exploratory study, but future directions are provided in the Discussion.

## 3. Method

For both experiments, sample size was determined based on a previous study on the foreign language effect in auditory modality [9].

### 3.1. Experiment 1

#### 3.1.1. Participants

Data from 184 (107 males, 77 females) Spanish native speakers from Spain were collected (mean age: 29.2 years, *SD*: 9.0 years). Participation was voluntary, the study was conducted in accordance with the Declaration of Helsinki, and the experimental protocol was approved by the Ethics Assessment Committee Humanities of Radboud University (reference number EAC 2018-9380).

#### 3.1.2. Materials

We used two classic moral dilemmas, i.e., the trolley dilemma and the footbridge dilemma [6,7], translated into Spanish (same texts as those used in [1], see Appendix A for all versions). In both dilemmas, a train has a problem and is going towards five people who will die if no action is taken. In the trolley dilemma, participants have the choice to divert the train to another track where one man is working. In the footbridge dilemma, participants are on a bridge and can choose to push a man on the track to stop the train. In both versions, one person will die if the action is taken. The dilemmas were recorded by a female native speaker of Spanish for the native accent (NA) condition. For the two foreign accent conditions, dilemmas were recorded in Spanish by a woman with a British English accent (FA-British), and a woman with a Cameroonian accent (FA-Cameroonian).

#### 3.1.3. Procedure

The experiment was presented online in Qualtrics (Qualtrics, Provo, UT). After receiving written instructions and giving their consent, participants were randomly assigned to one of three Accent conditions (NA, *N* = 62; FA-British, *N* = 60, or FA- Cameroonian, *N* = 62). They were asked to listen to the two dilemmas (order of presentation counterbalanced) and to make a yes/no decision to the questions ‘Would you change the track?’ (trolley dilemma) and ‘Would you push the man? (footbridge dilemma). A ‘yes’ answer to the dilemmas reflects utilitarian behaviour, whereas a ‘no’ answer reflects deontological behaviour. After indicating their decision, participants assessed the speaker for accent strength (‘How strong was the speaker’s accent?’ 1 = very mild, 7 = very strong) and comprehensibility (‘How difficult was it to understand the speaker?’ 1 = very difficult, 7 = not difficult), and completed questions about their demographic and language background. The experiment lasted about 5 min.

#### 3.1.4. Results

To check the hypothesis of whether a foreign accent affected moral judgement, we first looked at the decisions for the two dilemmas in each accent condition (see Table 1). A logistic regression model was conducted on the decisions (1 = yes, utilitarian vs. 0 = no, deontological) using the *glm* function in R (R Core Team, 2017, http://www.R-project.org/, accessed on 7 December 2021). Accent was treatment-coded with native accent as the reference level. Orthogonal sum-to-zero contrast coding was applied to dilemma type [38]. The footbridge dilemma was coded as −0.5 and the trolley dilemma as 0.5. The predictors and their interaction were entered into the model.

The results demonstrated an effect of dilemma type (*β* = 0.58, *SE* = 0.07, *t*-value = 8.10, *p* < 0.001), indicating that the odds of making a utilitarian decision for the trolley dilemma is higher than for the footbridge dilemma. No significant effect between the native accent and the British accent was found (*β* = 0.02, *SE* = 0.05, *t*-value = 0.48, *p* = 0.63). However, there was an effect between the native and the Cameroonian accent (*β* = 0.16, *SE* = 0.05, *t*-value = 3.18, *p* = 0.002), suggesting that the odds of making a utilitarian decision when the dilemmas were spoken in the Cameroonian accent is higher compared to the native accent. No interaction between dilemma type and accent was demonstrated (*p* > 0.1).

#### 3.1.5. Comprehensibility and Accent Strength

The results of accent assessment are presented in Table 2. A Kendall’s tau-b correlation demonstrated a moderately negative relation between accent strength and comprehensibility (τb = −0.41, *p* < 0.001), indicating that the stronger the accent the harder it is to understand it. Comprehensibility has been found to modulate decisions or the perception of the foreign speaker [29,30,39], hence, to explore whether it modulated the effect we observed across accent conditions, we ran two different tests. We first ran a logistic regression analysis to check whether comprehensibility influenced the strength or direction of the relationship between accent conditions and utilitarian responses. Accent was contrast-coded, with native accent coded as −0.5 and foreign accent coded as 0.5. The continuous predictor comprehensibility was centred around zero [40]. Both predictors and their interactions were entered into the model. The results demonstrated no significant effects for each predictor nor for their interaction (*p* > 0.1).

Secondly, we examined whether comprehensibility could be the underlying mechanism for the foreign accent effect on moral judgment, we therefore ran a mediation analysis using the *mediation* package in R [41] to check whether comprehensibility had a similar effect across all conditions. In this model, accent was contrast-coded and comprehensibility was uncentered. The results demonstrated that the effect of accent on moral judgment was not mediated by comprehensibility (*p* > 0.1).

### 3.2. Experiment 2

#### 3.2.1. Participants

A total of 251 (70 males, 181 females) Dutch native speakers from the Netherlands took part in the experiment (mean age: 28 years, *SD*: 12.9 years).

Participation was voluntary and the experimental protocol was approved by the Ethics Assessment Committee Humanities of Radboud University (reference number EAC 2018-9380).

#### 3.2.2. Materials

The same dilemmas as in Experiment 1 were used, except that they were recorded in Dutch. A female native speaker of Dutch recorded the texts for the native accent (NA) condition. For the foreign accent conditions, dilemmas were recorded by a woman with a British English accent (FA-British), a woman with a French accent from France (FA-French), and another woman with a Turkish accent from Turkey (FA-Turkish).

#### 3.2.3. Procedure

The procedure of Experiment 2 was identical as in Experiment 1.

#### 3.2.4. Results

To investigate whether a foreign accent affects moral judgement, we first examined the decisions for the two dilemmas in each accent condition (see Table 3). A similar analysis was conducted as in Experiment 1. A logistic regression analysis showed an effect of dilemma type (*β* = 0.71, *SE* = 0.07, *t*-value = 10.66, *p* < 0.001), indicating more utilitarian decisions for the trolley compared to the footbridge dilemma. More importantly, there was a significant interaction between dilemma type and the French accent (*β* = −0.25, *SE* = 0.10, *t*-value = −2.59, *p* < 0.01), indicating that the odds of making a utilitarian decision increased for the French accent compared to the native accent on the footbridge dilemma only. No significant effects were found for the British and the Turkish accent (*p* > 0.1).

#### 3.2.5. Accent Strength and Comprehensibility

The results of accent assessment are presented in Table 4. A Kendall’s tau-b correlation showed a moderately negative relation between accent strength and comprehensibility, (τb = −0.49, *p* < 0.001), indicating that the stronger the accent the harder to understand.

Next, we examined, as in Experiment 1, whether comprehensibility might contribute to the significant difference in moral judgments across native versus foreign accents. However, here we only focused on the data from the footbridge dilemma, as we had found a significant interaction between accent and dilemma type in the main analysis. Accent was contrast-coded, with native accent coded as −0.5 and foreign accent coded as 0.5. The continuous predictor comprehensibility was centred around zero [40]. Both predictors and their interaction were entered into the model. The results showed a significant interaction between accent and comprehensibility on moral judgment (*β* = 0.11, *SE* = 0.05, *t*-value = 2.12, *p* = 0.04). Unpacking this interaction revealed that comprehensibility played a significant role for the foreign accents (*β* = 0.11, *SE* = 0.05, *t*-value = 2.12, *p* = 0.04), but not for the native accent (*p* >0.1). However, unexpectedly, the odds of making a utilitarian increased when the comprehensibility of the foreign accents increased (see Figure 1).

Finally, we ran a mediation analysis on the data from the footbridge dilemma in which comprehensibility was uncentered and accent was contrast-coded. The results demonstrated that the effect of accent on moral judgment was not mediated by comprehensibility (*p* > 0.1).

In a final analysis, we examined whether we could find a foreign *accent* effect for all the speakers together (i.e., collapsing the Spanish and Dutch participants of Experiments 1 and 2). Accent and dilemma type were contrast-coded, with native accent and footbridge coded as −0.5 and foreign accent and trolley coded as 0.5. Both predictors and their interaction were entered into the model. The logistic regression results showed an effect of dilemma type (*β* = 0.62, *SE* = 0.03, *t*-value = 20.94, *p* < 0.001), demonstrating an increase in the odds of making utilitarian decisions on the trolley compared to the footbridge dilemma. More importantly, a significant effect between the native and the foreign accent was found (*β* = 0.07, *SE* = 0.03, *t*-value = 2.29, *p* = 0.02), indicating that the odds of making utilitarian decisions increase when listening to a foreign compared to a native accent.

## 4. General Discussion

This study tested the possibility of a foreign *accent* effect. In other words, we examined whether a foreign accent is a linguistic context that modulates (neuro)cognitive mechanisms and consequently impacts our behaviour. Our hypothesis originated from the foreign *language* effect, which refers to the fact that the language we use, native vs. foreign, modifies our decisions (e.g., [1,2,42]). The FLe has been attributed to a reduction in emotion, an increase in cognitive load, and psychological distance provoked by a foreign language [3]. Given that foreign-accented speech has been shown to affect these factors as well [15,25,29,30,31,32,33], we expected decisions to be modulated when processed in a foreign accent compared with a native accent. Indeed, when we presented participants with the Footbridge dilemma, we observed a significantly higher number of utilitarian responses in the foreign than in the native accent condition. This study is therefore the first observation of a foreign *accent* effect.

First of all, based on the FLe, we expected to observe the foreign *accent* effect in the footbridge dilemma (personal dilemma) but not in the trolley dilemma (impersonal dilemma). This pattern was true only in Experiment 2, but not in Experiment 1, and we do not have a clear explanation for the absence of interaction between dilemma type and accent in the latter. We expected such a pattern because personal dilemmas involve a larger amount of emotionality than impersonal dilemmas [4], and given that foreign-accented speech reduces emotionality [15], it leads to an increase in utilitarian responses when emotionality is high. The same mechanism has been proposed to account for the FLe [1,42,43] and it is consistent with the dual-process account which claims that decision making is an interplay between controlled processes and emotional processes [4,5]. Note that alternatives to the emotion reduction account have been proposed to explain the FLe, such as a reduced importance of the consequences [44] and reduced concern for causing harm [45]. Our study was not designed to disentangle these alternatives, which cannot be excluded. Further research should investigate the impact of foreign-accented speech on the perception of consequences and potential harm caused by one’s actions.

It is important to underline that the effect was observed both with native speakers of Spanish and native speakers of Dutch, which suggests that the effect is not language- or culture-dependent. However, it was not observed with all the foreign accents we used. Several explanations are possible to account for these results. The first explanation is cognitive fluency. Further analyses of the responses obtained for the footbridge dilemma in Experiment 2 revealed a relationship between comprehensibility and the effect of accent (no such effect was found in Experiment1). However, surprisingly, we observed that a decrease in the comprehensibility of a foreign accent resulted in a decrease in utilitarian decisions. This explanation is in line with our original hypothesis that the cognitive load generated when processing foreign-accented speech affects moral judgements, but it is reversed. Indeed, we expected an increase in utilitarian responses as comprehensibility decreased. Nevertheless, independently of the direction of the effect, our results suggest that comprehensibility modulates moral judgements, which is consistent with previous studies that have shown an effect of cognitive fluency on decision making in other contexts, such as when deciding whether a statement is true or not [29] or whether to buy one product or another [37]. Hence, the fact that the foreign *accent* effect was not observed with all the accents may depend on the level of comprehensibility of each speaker. Here, the perception of each speaker in relation to comprehensibility was assessed by each participant after answering the dilemma; future research should investigate the role of accent fluency on decision making by including these factors as experimental conditions (e.g., mild or strong accent, as in Lev-Ari & Keysar, 2010, for example).

The second potential explanation (not exclusive of the previous one) is the social categorisation of the speaker. As mentioned in the Introduction, an accent reveals the foreignness of a speaker, who is immediately categorised as a member of a different social group. This categorisation provokes psychological distance, which may lead individuals to consider situations in a more abstract way, and, in the context of moral judgements, could increase utilitarian responses [34,35]. The psychological distance may depend on the closeness between the native speaker’s and the foreign speaker’s social group. Indeed, social categorisation may trigger stereotypes associated with the speaker’s culture [36,37], which may be positive or negative. For instance, a study has revealed that native Dutch people would prefer to spend time with north-European migrants (such as the French) above south-Europeans (Italians), both of these being preferred over ex-colonial groups such as the Surinamese people, and at the bottom of this hierarchy are people from North-African or Middle-Eastern descent, such as Turkish people [46]. The influence of culture has already been shown to modulate the FLe [47]. However, our results do not entirely support the stereotype explanation given that we observed a significant effect with the French accent but not with the Turkish accent when the reverse would be expected. The impact of social categorisation on moral judgements should be further investigated by manipulating the stereotypes triggered by an accent as experimental conditions (i.e., positive vs. negative).

Finally, because we manipulated accents, the dilemmas were presented in auditory modality. Previous studies that have looked at the effect of modality (written vs. auditory) on the FLe [8,9,10]. Muda and collaborators have suggested that modality of observation does not seem to explain the variability observed in the FLe, however, in their study they used a text-to-speech system to generate the audio dilemmas. Here, since we used real speakers, we cannot exclude that their idiosyncratic features may have affected participants’ language attitudes towards them, and consequently, their decisions, which could explain the variability in the presence of an effect across accents. Further studies looking at the effect of a speaker’s individual characteristics on moral judgements (or other cognitive process) in auditory modality is needed. The findings would also have experimental implications for studies that use this modality with other methodologies (e.g., eye-tracking, event-related potentials, virtual reality, among others). Note, however, that here, when we analysed all the speakers together (collapsing the two experiments), the variability across speakers disappeared and we observed an overall significant increase in utilitarian responses in the foreign condition compared with the native condition. Hence, one option to avoid effect of speakers’ idiosyncratic features may be to use various speakers within a same condition.

This study has some limitations. First, it was conducted online so the researchers had less control over the participants. Second, we only used the trolley and footbridge dilemmas, which are sacrificial dilemmas, and concerns about the validity of this type of dilemmas have been raised [48,49]. Given that a recent study has revealed no effect of foreign accent on decisions regarding social norms [50], the robustness of the effect in other contexts should be tested using different, more ecologically valid dilemmas. Finally, as an anonymous reviewer pointed out, there is a possibility that negative opinions of or discrimination against obese people may have interacted with native or foreign cultures for the footbridge dilemma. The interaction between a foreign accent effect on moral judgements and cultural biases should be investigated in future research.

To conclude, this study is the first demonstration of a foreign *accent* effect on moral judgements. Although we observed variability in the effects across accents that cannot be explained with the present design, the findings suggest that a foreign accent, like a foreign language, is a linguistic context that modulates (neuro)cognitive mechanisms and consequently impacts our behaviour. More research is needed to follow up on this exploratory study and understand the influence of factors such as emotion reduction, cognitive load, psychological distance, and speaker’s idiosyncratic features on moral judgments.

## Figures and Tables

**Figure 1 brainsci-11-01631-f001:**
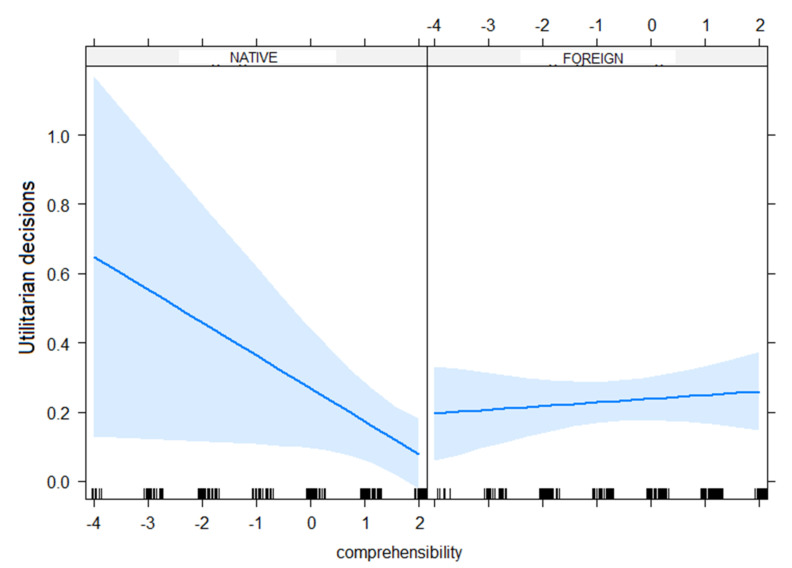
Interaction between accent (native versus foreign) and comprehensibility (‘How difficult was it to understand the speaker?’ 1 = very difficult, 7 = not difficult) on moral judgment (utilitarian decisions) for the Dutch participants (Experiment 2).

**Table 1 brainsci-11-01631-t001:** Percent of utilitarian (‘yes’) decisions for the two dilemmas in the native and the two accent conditions in Experiment 1.

	Footbridge	Trolley
NA (N = 62)	19	77
FA-British (N = 60)	17	85
FA-Cameroonian (N = 62)	40	89

**Table 2 brainsci-11-01631-t002:** Results of accent assessment in the native and the two accent conditions in Experiment 1 for accent strength (‘How strong was the speaker’s accent?’ 1 = very mild, 7 = very strong) and comprehensibility (‘How difficult was it to understand the speaker?’ 1 = very difficult, 7 = not difficult). Standard deviations are displayed in parentheses.

	Accent Strength	Comprehensibility
NA (N = 62)	2.2 (1.4)	6.5 (1.5)
FA-British (N = 60)	5.4 (1.0)	6.3 (1.1)
FA-Cameroonian (N = 62)	5.9 (1.1)	3.5 (1.5)

**Table 3 brainsci-11-01631-t003:** Percent of utilitarian (‘yes’) decisions for the two dilemmas in the native and the three accent conditions in Experiment 2.

	Footbridge	Trolley
NA (N = 69)	12	83
FA-British (N = 63)	19	78
FA-French (N = 63)	32	78
FA-Turkish (N = 56)	18	89

**Table 4 brainsci-11-01631-t004:** Results of accent assessment in the native and the three accent conditions in Experiment 2 for accent strength (‘How strong was the speaker’s accent?’ 1 = very mild, 7 = very strong) and comprehensibility (‘How difficult was it to understand the speaker?’ 1 = very difficult, 7 = not difficult). Standard deviations are displayed in parentheses.

	Accent Strength	Comprehensibility
NA	2.0 (1.3)	6.5 (1.0)
FA-British	5.8 (1.1)	4.8 (1.4)
FA-French	5.8 (1.1)	3.2 (1.3)
FA-Turkish	5.1 (1.3)	4.9 (1.4)

## Data Availability

The data presented in this study are openly available in OSF at https://osf.io/rey5w/ (accessed on 7 December 2021).

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
