# Peer review of "Is There a Foreign Accent Effect on Moral Judgment?"

_brainsci, 2021, doi:10.3390/brainsci11121631_

Round 1

Reviewer 1 Report

The paper tests an interesting hypothesis – does processing information in provided in foreign accent influence moral decision making in the same manner as processing it in foreign language? This question is interesting and could have both theoretical and real-world implications. It would be of interest to readers and I would like to see it published. That said, the authors made several surprising choices with regards to how they analyzed the data. The experiments should be reported and analyzed more in-line with the field’s norms and with the theoretical claims.

  • The authors predict that foreign accent should affect decisions in the footbridge dilemma but not in the trolley dilemma. In other words, the authors predict an interaction between accent and dilemma type yet they do not statistically test for such an interaction. Instead, they analyze each dilemma type separately. Finding an effect in one dilemma and not in another one is not sufficient to say that the dilemmas are different from each other. The authors should conduct a single model in each experiment and report the results of the interaction in addition to the simple effects they report.
  • The authors divide each experiment into 2, yet it seems that they reuse the same data across Experiments 1a and 1b (and presumably across 2a and 2b but they don’t provide enough information to be certain). It seems like they tested the native condition only once but each time analyzed the same data in contrast to another condition. This is quite odd. As the experiment and analyses reuse the same native data, they should combine experiments a and b and analyze and report them in the same analysis.
  • The authors test whether the effect of foreign accent is due to disfluency. They therefore measure comprehensibility and accent strength. They then run an analysis with the interaction of each of these with accent (if I understood correctly). The authors might want to reconsider how to run these analyses.
    (1) If the hypothesis that is tested is that disfluency is the underlying mechanism, then the authors should run a mediation test. That is, the analysis of an interaction is better suited for testing moderation, not mediation. A moderation would be that comprehensibility would have a different effect in different accent conditions, but if disfluency is the underlying mechanism, disfluency would have the same effect across all conditions. That is, it is not that disfluency would only matter sometimes, but that disfluency should vary across conditions and therefore lead to different decisions. In the case of a native accent, disluency should simply be low and thus lead to different decisions from the accent conditions. Indeed, the authors never explain why would anyone expect the opposite effect in the native condition (see also below). The authors might want to use one of several mediation packages in R (e.g., Tingley D, Yamamoto T, Hirose K, Keele L, Imai K (2014). “mediation: R Package for Causal Mediation Analysis.” Journal of Statistical Software, 59(5), 1–38).
    (2) The authors report that comprehensibility increased utilitarian decisions in the native conditions but decreased it in the other conditions. The author, however, did not report the simple effects. Therefore, it is unclear whether comprehensibility significantly influenced performance in any of the conditions but only that its effect differed across conditions. For example, the error bands are very wide for low comprehensibility in the native condition so it’s very likely that this simple effect would not be significant (or driven by 1-2 outlier participants who surprisingly struggled with the native speaker). To add to (1), an interaction is probably not the best way to analyze this, as the different conditions don’t even have the same range of comprehensibility ratings.   
    (3) Is accent strength correlated with accent comprehensibility? If so, entering them into the analysis at the same time means that all the shared variance in excluded, thus reducing their effects. The authors might also want to explore whether this is the case and when they enter only one of them there is an effect.
    (4) It is not fully clear what is the hypothesis regarding accent strength and why it is measured and analyzed. Accent strength, if independent of comprehensibility, should not influence disfluency. As accent strength is also theoretically independent of culture (speakers of the same language can have mild or heavy accents), it also does not represent psychological distance. The authors should explain why they expect it to influence performance.
  • The authors should explain how they decided on sample size.
  • The title for the y-axis in Figure 1 should be more informative. In general, the plot is confusing as it doesn't seem to fit with the way the data were reported beforehand. The y-axis states ‘moral decision’, so it’s unclear whether it codes utilitarian or deontological decisions. Earlier in the text, it was stated that utilitarian decisions were coded as 1 and deontological as 0, so higher values should be more utilitarian decisions. If that is the case though, the results are the opposite of what was reported.

Reviewer 2 Report

The authors of the reported study attempt to find evidence of the foreign accent effect, in addition to the foreign language effect. They do so by having a substantial number of participants from 2 different cultures/L1 backgrounds listen to two dilemmas, one known to be affected by emotionality, in their native language accent and in two other language accents. They claim to have found evidence of the dilemmas eliciting more utilitarian responses when presented in the foreign accent for the more emotionally charged dilemma, albeit not in every native language x accent combination. 

I enjoyed reading the study, the introduction was sufficiently detailed, methods mostly clear, and results well presented. The conclusions did not overreach beyond what results could provide evidence for. The study provides novel, interesting findings, with real-life significance.

I have just a few comments which authors and editors might find useful.

  1. I am not sure what the reasoning behind this choice was, but I would absolutely refrain from using 'African' as the Cameroonian accent descriptor, in the abstract or otherwise. Country names should be used to mark accents, as was done with Turkish and French, i.e. Cameroonian should be used instead. (Listing the language(s) the individual speaks would be best, but it might not be practical due to presumed multilingualism in Cameroon.)
  2. Given the recent push for transparency in research, will the authors provide the data and analysis pipeline in a repository? If I missed the link for it, I apologize.
  3. In some cases, the model structure was not completely clear, i.e. what the main effects were and interactions explored. For example, I would make it more explicit that all accent levels (native, foreign 1, foreign 2) were included in the accent strength and comprehensibility analysis.
  4. I'm also curious why random effects were not included. 
  5. Related to the previous points, model outputs would be very useful in the supplementary materials, or in an accessible repository.
  6. I would personally not mention the marginal effect in line 202 (p = 0.076), as it is not that close to 0.05.
  7. I'm curious whether the additional layer of negative opinions of or discrimination against obese people could have interacted with native or foreign cultures for the Footbridge dilemma. 
  8. In line 119, you say that "[you] expected participants to be more likely to kill one person to save five when facing [...]". I would rephrase this to "expected participants to be more likely to state they would kill one person to save five".
